# Enhancing EQ-5D-5L Sensitivity in Capturing the Most Common Symptoms in Post-COVID-19 Patients: An Exploratory Cross-Sectional Study with a Focus on Fatigue, Memory/Concentration Problems and Dyspnea Dimensions

**DOI:** 10.3390/ijerph21050591

**Published:** 2024-05-03

**Authors:** Helena Janols, Carl Wadsten, Christoffer Forssell, Elena Raffeti, Christer Janson, Xingwu Zhou, Marta A Kisiel

**Affiliations:** 1Department of Medical Sciences, Section of Infectious Diseases, Uppsala University, 751 85 Uppsala, Sweden; 2Department of Statistics, Uppsala University, 751 20 Uppsala, Sweden; callewadsten@gmail.com; 3Department of Medical Sciences, Uppsala University, 751 85 Uppsala, Sweden; christoffer.forssell.1408@student.uu.se; 4Department of Global Public Health, Karolinska Institute, 171 77 Stockholm, Sweden; elena.raffetti@geo.uu.se; 5Department of Medical Sciences, Respiratory, Allergy and Sleep Research, Uppsala University, 751 85 Uppsala, Sweden; christer.janson@medsci.uu.se; 6Department of Medical Sciences, Clinical Physiology, Uppsala University, 751 85 Uppsala, Sweden; 7Department of Medical Sciences, Occupational and Environmental Medicine, Uppsala University, 753 10 Uppsala, Sweden

**Keywords:** EQ-5D-5L, fatigue, memory/concentration problems, dyspnea, EQ-VAS, COVID-19

## Abstract

This study aimed to determine whether the EQ-5D-5L tool captures the most common persistent symptoms, such as fatigue, memory/concentration problems and dyspnea, in patients with post-COVID-19 conditions while also investigating if adding these symptoms improves the explained variance of the health-related quality of life (HRQoL). In this exploratory cross-sectional study, two cohorts of Swedish patients (n = 177) with a history of COVID-19 infection answered a questionnaire covering sociodemographic characteristics and clinical factors, and their HRQoL was assessed using EQ-5D-5L with the Visual Analogue Scale (EQ-VAS). Spearman rank correlation and multiple regression analyses were employed to investigate the extent to which the most common persistent symptoms, such as fatigue, memory/concentration problems and dyspnea, were explained by the EQ-5D-5L. The explanatory power of EQ-5D-5L for EQ-VAS was also analyzed, both with and without including symptom(s). We found that the EQ-5D-5L dimensions partly captured fatigue and memory/concentration problems but performed poorly in regard to capturing dyspnea. Specifically, the EQ-5D-5L explained 55% of the variance in memory/concentration problems, 47% in regard to fatigue and only 14% in regard to dyspnea. Adding fatigue to the EQ-5D-5L increased the explained variance of the EQ-VAS by 5.7%, while adding memory/concentration problems and dyspnea had a comparatively smaller impact on the explained variance. Our study highlights the EQ-5D-5L’s strength in capturing fatigue and memory/concentration problems in post-COVID-19 patients. However, it also underscores the challenges in assessing dyspnea in this group. Fatigue emerged as a notably influential symptom, significantly enhancing the EQ-5D-5L’s predictive ability for these patients’ EQ-VAS scores.

## 1. Introduction

Coronavirus disease 2019 (COVID-19), caused by severe acute respiratory syndrome coronavirus 2 (SARS-CoV-2) infection, has led to a significant increase in global morbidity and mortality [1]. Additionally, a subset of infected patients experience persistent symptoms that may last for at least three months following the infection, which is referred to as post-COVID-19 condition by the World Health Organization (WHO) [2]. Although the prevalence of post-COVID-19 conditions has yet to be determined [3], the WHO estimates that around 10% of people might suffer from this condition following COVID-19 infection [4]. 

Post-COVID-19 conditions of varying severity can occur after COVID-19 and present a range of persistent symptoms, with fatigue, dyspnea and cognitive problems being the most common [1,3]. These enduring symptoms can hamper daily activities, limit physical activity, lead to psychological symptoms and cognitive dysfunction, and subsequently impact quality of life [5,6,7,8]. Currently, there is no standardized clinical assessment for measuring the impact of post-COVID-19 on an individual’s everyday life [9].

The EQ-5D-5L is a widely used and concise tool for assessing generic health-related quality of life (HRQoL) [10]. It consists of self-assessment questions that cover five health dimensions: mobility, self-care, usual activities, pain/discomfort, and depression/anxiety. Each dimension offers five possible response options, ranging from ‘no problems’ to ‘extreme problems’ [11]. The EQ-5D-5L has been extensively useful in cost-utility analysis, aiding policy-makers in allocating healthcare resources [12]. Recently, it has been applied in studies examining the outcomes of post-COVID-19 patients [13,14]. However, concerns have been raised about its brevity, limiting its ability to measure the long-term consequences of infectious diseases such as COVID-19 [15]. A previous study on patients with persistent symptoms following another infectious disease, Q-fever, revealed that EQ-5D-5L could capture fatigue and cognitive problems related to that condition [16]. To date, no studies have explored whether EQ-5D-5L dimensions and the EQ visual analogue scale (EQ-VAS) can adequately capture significant symptoms in post-COVID-19 conditions. 

This exploratory study aimed to investigate whether EQ-5D-5L dimensions capture the most common persistent symptoms, such as fatigue, memory/concentration problems and dyspnea, in patients with post-COVID-19 conditions and whether adding these symptoms improve the explained variance of HRQoL. 

## 2. Materials and Methods

### 2.1. The Study Design and Cohorts

This study is a component of the longitudinal project at the Uppsala University Hospital, known as “COMBAT post-COVID”, which investigates the long-term consequences of COVID-19 [17,18,19]. This exploratory cross-sectional study comprised the following two cohorts of adults (age 18 years and older) who had a history of COVID-19 between 2020 and 2022: “Hospitalized COVID” and “Post COVID outpatients”. The data included in this study were gathered through a questionnaire. 

The “Hospitalized COVID” cohort included patients who had been admitted to the Department of Infectious Diseases at the Uppsala University Hospital for COVID-19 treatment. The patients had confirmed COVID-19 infection through a positive polymerase chain reaction (PCR) test for SARS-CoV-2 in a nasopharyngeal swab. They were contacted by a telephone call between ten and eleven months after their initial infection. Furthermore, between April and July 2021, twelve months after the initial infection, 57 of the 149 hospitalized patients (38%) participated in a COVID-19 follow-up visit at the Department of Respiratory, Allergy and Sleep Research at the Uppsala University Hospital (Figure 1). 

The “Post COVID outpatients” cohort comprised patients who were referred to the post COVID-19 outpatient clinic in the Uppsala Region. Enrolled criteria for the outpatients included the presence of persistent symptoms lasting at least 12 weeks after a COVID-19 diagnosis, which could be microbiologically verified by a positive PCR for SARS-CoV-2 in a nasopharyngeal swab or probable, as per the WHO and Delphi study, referred to as post COVID-19 condition [2]. Between October 2021 and December 2022, a survey was sent to the home addresses of all enrolled patients (n = 198) in the post-COVID-19 cohort, along with a return envelope. A single reminder was sent to all participants within one month. Out of the invited patients, 123 (62%) responded (Figure 1). 

From the two cohorts, a total of 180 patients were initially included. Three individuals had missing answers within the EQ-5D-5L and were excluded from the analyses. The final study population comprised 177 patients (Figure 1). 

### 2.2. EQ-5D-5L

Patients assessed their HRQoL using the EQ-5D-5L, which consisted of five dimensions: mobility, self-care (ADL), usual activities, pain/discomfort and anxiety/depression. Each dimension was evaluated for severity using the following five-point scale: 1 = no problem, 2 = slight problems, 3 = moderate problems, 4 = severe problems and 5 = extreme problems [11]. Additionally, the questionnaire included a Visual Analogue Scale (EQ-VAS) where responders rated their health on a vertical axis ranging from zero, “the worst imaginable condition”, to 10, indicating “the best imaginable condition”. The EQ-5D-5L was present as a health profile by combining the responses into a five-number profile ranging from 11111 (full health) to 55555 (worst health) [11].

### 2.3. Persistent Symptoms and Their Severity

Patients were asked about any persistent symptoms following their initial COVID-19 infection. These symptoms included problems with fatigue, memory/concentration, dyspnea, cough, sore throat, nasal congestion, impaired taste and/or smell, heart palpitations, chest pain, vertigo, headache, muscle/joint pain, sleeping, depression, anxiety, gastrointestinal tract (including nausea, vomiting and stomach pain) and skin. Patients were then asked to rate the intensity of each symptom on a scale from 1 (very mild) to 10 (most severe). A score of zero was assigned if the patient did not experience any symptoms. The most common persistent symptoms were problems with fatigue, memory/concentration and dyspnea (Appendix A). These symptoms were subsequently used in further analyses. Based on previous studies on pain, fatigue, and dyspnea graded on a 10-point scale, we set the optimal cutoff point to distinguish between no/mild and severe symptom severity at 7 [20,21]. 

### 2.4. Covariates

The questionnaire collected sociodemographic characteristics, including education level (classified as at least three years of university education resulting in an academic degree, at least two years vocational school, and up to secondary school), marital status (categorized as married, living with a partner, single, divorced, widowed), country of birth (categorized as Sweden or other countries), primary work status (categorized as working, unemployed, on sick leave, retired, student or other), smoking status (classified as never, ex or current smoker) and use of snuff (user or non-user). Age and sex at birth (females or males) were determined using the patient’s Swedish personal identification number. 

Participants were also inquired about the severity of COVID-19 symptoms at onset (including very mild/mild, moderate, severe, or very severe) and whether they were hospitalized at infection onset. They reported any pre-existing medical conditions diagnosed by a doctor, including hypertension, heart disease (such as heart failure or acute myocardial infarction), thyroid disease, diabetes mellitus, lung disease (including asthma and chronic obstructive pulmonary disease (COPD)), cancer, conditions requiring immunosuppressive treatment, depression/anxiety and chronic pain. Participants provided their weight and height, which were then used to calculate their body mass index (BMI). 

### 2.5. Ethics

This study followed the Helsinki Declaration and was approved by the Swedish Ethical Review Authority (Dnr 2021-01891 and Dnr 2022-01261-01). All participants gave written informed consent to use their questionnaire answers. 

### 2.6. Statistical Analysis 

Descriptive statistics were used to present all patients’ sociodemographic and clinical characteristics. Categorical variables were expressed as proportions, continuous variables as means with standard deviations, as well as ordinal variables as medians with interquartile ranges (IQR). Missing values within sociodemographic and clinical variables were <5% across all categorical variables and were not included in percentage calculations (Appendix A).

Patients with the most common persistent symptoms, fatigue, memory/concentration problems and dyspnea, were divided into two severity groups based on the rating score on a 10-point scale, with a cutoff at 7. 

The EQ-5D-5L health states were converted into a single utility score using a scoring algorithm. This study used the Swedish value set and scoring algorithm to calculate utility scores [22]. The potential values for Sweden from this algorithm ranged between −0.314 (worst imaginable health status) and 1 (best imaginable health status) [22]. 

Spearman’s correlation was used to investigate correlations among fatigue, memory/concentration problems, dyspnea, EQ-5D-5L dimensions and the utility score. Correlation coefficients were interpreted as follows: 0.1–0.29 (poor), 0.3–0.5 (fair), 0.6–0.79 (moderately strong) and 0.8–1.0 (very strong) [23].

Variance Inflation Factor (VIF) values were calculated to assess potential multicollinearity between different EQ-5D-5L dimensions. A VIF of less than 5 indicated a low correlation of that predictor compared to other predictors [24]. We found a low VIF value that excluded significant multicollinearity concerns in the subsequent statistical models (Appendix A). 

Multiple regression analysis was used to assess the impact of EQ-5D-5L dimension scores (independent variables) on the severity of the symptoms (fatigue, memory/concentration problems and dyspnea (dependent variables)). An exploratory analysis involving nine distinct multiple regression models to compare adjusted R-squared values for EQ-VAS (dependent variable) was always undertaken. 

All statistical analyses were conducted using R, version 4.1.1 (R Core Team, Vienna, Austria, 2023) [25]. The statistical significance level was set as 0.05.

## 3. Results

### 3.1. Study Population and Health Outcomes 

This study included 177 patients, of whom 55.4% were female, and the average age of the study participants was 52 years (Table 1). The mean EQ-5D-5L utility score in the study population was 0.77 (Table 1). All subjects who reported no problems on all EQ-5D-5L dimensions were categorized into the no/milder symptoms group, except for one with severe fatigue problems. Perfect health on the EQ-5D-5L was reported by only 7% of the study patients. Patients in the higher severity groups of symptoms exhibited a lower prevalence of current employment, along with a higher incidence of comorbidities (such as heart disease, lung disease, depression/anxiety and chronic pain). They also reported a higher number of persistent symptoms following COVID-19 and a lower EQ-VAS score, indicating the worst self-reported HRQoL (Table 1). For example, patients experiencing severe fatigue reported an average of 3.6 persistent symptoms and an EQ-VAS score of 3. In contrast, those in the no/mild fatigue group had an average of 1.2 persistent symptoms and an EQ-VAS score of 6.

### 3.2. Distribution of Patients on EQ-5D-5L Dimensions for Fatigue, Memory/Concentration Problems and Dyspnea

The proportion of study patients reporting problems on the EQ-5D-5L was higher when experiencing severe fatigue, memory/concentration problems or dyspnea in all dimensions. Among patients with severe fatigue, memory/concentration problems and dyspnea, a notably higher proportion reported severe or extreme problems, particularly in the usual activities and pain/discomfort dimensions. For example, a substantial percentage of patients with severe fatigue, memory/concentration problems and dyspnea reported extreme problems in “usual activities”. Notably, the self-care dimension exhibited the fewest reported problems across all respondents (Figure 2).

### 3.3. Correlation of EQ-5D-5L Dimensions and Utility Scores with Fatigue, Memory/Concentration Problems, and Dyspnea

The strongest correlations were found between fatigue and memory/concentration problems and between fatigue or memory/concentration problems and the “usual activities” dimension of EQ-5D-5L (Table 2 and Appendix A). Moderate correlations were observed between fatigue and memory/concentration problems and the other four dimensions of EQ-5D-5L (i.e., mobility, self-care, pain, and anxiety). Both fatigue and memory/concentration problems had a strong negative correlation with the utility scores, while dyspnea had a moderate negative correlation with the utility scores.

### 3.4. Multiple Regression Analyses

Multiple regression analyses showed that the EQ-5D-5L dimensions explained 47.1% of the variance in fatigue, 54.6% in memory/concentration problems and 14.4% in dyspnea (Table 3). For fatigue and memory/concentration problems, it was evident that reporting problems at any level of “usual activities” significantly increased the severity of these symptoms compared to having no problems with “usual activities”. A similar pattern was found for experiencing severe problems with “self-care” (level 4) compared to no problems with self-care. No study patient reported an extreme problem with “self-care” (level 5). In addition, for memory/concentration problems, we found that reporting slight to moderate problems with anxiety/depression was linked to more severe complaints related to these symptoms compared to not having anxiety/depression. In the context of dyspnea, reporting slight problems with “self-care” (level 2) was associated with more pronounced complaints related to dyspnea compared to having no problems in this dimension (level 1). Furthermore, study patients who scored “1” (level 1) across all EQ-5D-5L dimensions reported fatigue, memory/concentration problems and dyspnea as 1.76, 0.99 and 1.48, respectively, on a scale of 0−10.

### 3.5. Explanatory Power of EQ-5D-5L with and without Symptoms for EQ-VAS 

The exploratory power of the EQ-VAS for the EQ-5D-5L’s utility score was lower than for the EQ-5D-5L dimensions (37.6% vs. 57.7%, Table 4). When comparing the explained variance of the EQ-VAS for EQ-5D-5L dimensions with symptom(s), we found that adding fatigue increased the exploratory power the most, with an increase of 5.5%. Then, adding memory/concentration problems or dyspnea increased the exploratory power by 1% and 0.8%, respectively. Adding fatigue in combination with memory/concentration problems or dyspnea to the EQ-5D-5L resulted in a 5.3% increase in exploratory power, whereas adding all symptoms showed a 5.1% increase. 

## 4. Discussion

### 4.1. Main Findings 

Our study revealed that, while the EQ-5D-5L dimensions offered some insight into fatigue and memory/concentration problems, they performed poorly in capturing dyspnea. Specifically, the EQ-5D-5L explained 55% of the variance in memory/concentration problems, 47% in terms of fatigue and only 14% in terms of dyspnea. Among these dimensions, the “usual activities” dimension exhibited the strongest correlation with fatigue and memory/concentration problems. This correlation was consistent across all levels within this dimension, indicating a progressive increase in problems with usual activity corresponding to the severity of fatigue and memory/concentration problems. In contrast, the other EQ-5D-5L dimensions demonstrated only moderate to weak correlations with fatigue and memory/concentration problems. Notably, the “usual activities” dimension alone showed a stronger association with fatigue and memory/concentration problems than the utility score representing all EQ-5D-5L dimensions. 

Additionally, we found that the “self-care” dimension had a statistically significant impact on fatigue and memory/concentration problems; however, this was only at level 4 within this dimension. Notably, no patient reported level 5 in the “ADL” dimension. This observation suggests that the study patients generally did not experience high levels of problems in their daily living activities.

Our study revealed that dyspnea displayed weak associations with all EQ-5D-5L dimensions, and these correlations were inconsistent across different levels of any dimension. When assessing dyspnea, we did not identify a statistically significant trend across multiple levels of any EQ-5D-5L dimension. Instead, we found only an isolated significant association with slight problems in the “self-care” dimension. However, our descriptive analysis has highlighted a higher prevalence of moderate, severe and extreme problems in the “mobility” dimension, as well as severe and extreme problems in the “usual activities” dimension and “depression and anxiety” dimension among patients with severe dyspnea in comparison to those with no/milder symptoms. This suggests that there might be unaccounted factors or complexities underlying dyspnea in patients with post-COVID-19 conditions that may contribute to the weak overall associations observed in the regression analysis. Further research is needed to explore these associations.

In our exploratory analysis, we observed that adding fatigue to the EQ-5D-5L significantly improved the explained variance of the EQ-VAS; however, adding memory/concentration problems or dyspnea had little effect on the explained variance. Notably, when two symptoms, such as fatigue and memory/concentration problems or fatigue and dyspnea, were added to the EQ-5D-5L, the explained variance was slightly lower than when adding fatigue alone. This may be attributed to the strong correlation between fatigue and memory/concentration problems and the moderate correlation between fatigue and dyspnea. 

We found that using the utility score as an independent variable alone resulted in significantly weaker explained variance compared to using all the EQ-5D-5L dimensions as independent variables. This suggests that, despite the utility score being a summary of the dimensions, a substantial amount of information is lost when using it exclusively. Therefore, our finding suggests that it may be more informative in clinical practice to consider all the EQ-5D-5L dimensions rather than relying solely upon the utility score that represents them. 

### 4.2. Comparison to Previous Studies 

Our findings were consistent with previous research by Geraerds et al., finding that EQ-5D-5L partially captures fatigue and memory/concentration problems in patients with Q-fever [16]. Similarly, a study on COPD patients observed a correlation between EQ-5D-5L and fatigue [26]. A Dutch study found that, in the general population, fatigue was partially covered by the EQ-5D-5L with the domains “usual activities” and “pain and discomfort”, whereas “self-care” contributed the least. In this study, the link between EQ-5D-5L domains and fatigue was stronger in subjects with at least one chronic disease than healthy individuals [27]. However, a study found limited additional explanatory power of EQ-5D-5L with regard to fatigue [15]. This disparity might arise from their use of the three-level EQ-5D variant and the simultaneous inclusion of multiple dimensions, which could have weakened the impact of individual symptoms. 

Furthermore, the cognitive dimension has been added to EQ-5D-5L and EQ-5D in studies involving patients following stroke or hearing and vision impairments [28,29]. In trauma-related research, the addition of questions about cognitive symptoms has been shown to enhance the EQ-5D’s explanatory power for EQ-VAS [30,31,32]. 

In terms of respiratory symptoms, our results align with the findings from a study on COPD patients that suggested a potential inadequacy in the EQ-5D-5L’s ability to capture dyspnea, especially in a generally healthy population [33]. However, in contrast to our study, Nolan et al. found a strong correlation between the utility score and dyspnea while utilizing a disease-specific assessment for chronic respiratory conditions [26]. This discrepancy raises the possibility of enhancing the EQ-5D-5L by including questions related to respiratory symptoms.

Finally, it is important to note that, in our study, fatigue significantly surpassed memory/concentration problems, contributing to the EQ-5D-5L’s predictive capacity for EQ-VAS scores. This nuance challenges our findings in the context of the post-COVID-19 condition. Nonetheless, the potential added value of incorporating a fatigue dimension into the EQ-5D-5L should be explored using various methodologies in future studies on the sequelae of infectious diseases such as COVID-19.

### 4.3. Strengths and Limitations

The strength of this study is that it is among the first to evaluate the sensitivity of the EQ-5D-5L in capturing fatigue, memory/concentration problems and dyspnea in patients with post-COVID-19 conditions. Furthermore, it includes two cohorts of Swedish patients with different levels of severity of initial COVID-19 infection, both hospitalized and non-hospitalized. This diversity provides a more comprehensive understanding of the HRQoL and the severity of persistent symptoms in patients with post-COVID-19. Additionally, the measurement of the HRQoL employed the EQ-5D-5L and the EQ-VAS, both robust and comprehensive tolls. These instruments have been previously validated across diverse populations.

The study carries some potential limitations. Firstly, the relatively small number of participants and modest response rate may impede the generalizability of our results. Therefore, it is important to view this study as exploratory in nature. Secondly, the representativeness of the study sample may be constrained due to testing limitations in Sweden during the early days of the pandemic. A notable part of the cohort from the post-COVID-19 outpatients did not have a laboratory-confirmed diagnosis, further limiting the study’s scope and generalizability. 

Third, a limitation arises from the absence of data regarding the duration of persistent symptoms, as this information was not available during the study for post-COVID outpatients. However, all patients reported symptoms being persistent for at least three months, meeting the criteria for the post-COVID-19 condition diagnosis [2]. 

Fourth, the methodology employed a simple question to assess persistent symptoms and rate their severity on a 10-point scale, with a cutoff of 7, drawing inspiration from other scales designed for dyspnea and pain. Notably, a relatively small proportion of patients reported severe dyspnea, potentially introducing limitations to the precision of the measurements for this specific symptom (defined as differential misclassification). On the contrary, the number of patients in the subgroups categorized as having no/mild or severe severity levels for the symptoms of fatigue and memory/concentration problems was approximately equal. 

Additionally, some information collected through the questionnaire, such as severity of symptoms at infection onset, may introduce recall bias, as patients might have difficulty accurately recalling past details. Another limitation of our study is the lack of information on vaccination, which might be an important variable, as vaccines generally reduce the risk of severe COVID-19, reinfection and its consequences in regard to post-COVID-19 [34]. 

## 5. Conclusions 

Our exploratory study has illuminated the strengths and challenges of the EQ-5D-5L tool in assessing HRQoL among post-COVID-19 patients. The EQ-5D-5L demonstrated its partial ability to capture fatigue and memory/concentration problems in these patients. However, it also faced difficulties in adequately addressing dyspnea. The addition of a fatigue dimension emerged as a valuable enhancement in the context of post-COVID-19 conditions. This could enhance the tool’s sensitivity and ability to capture these patients’ everyday life problems, and it has the potential to identify those who are most affected by their symptoms, enabling healthcare providers to prioritize them for rehabilitation efforts. We strongly recommend more studies to improve the EQ-5D-5L in order to better assess the multidimensional effects of post-COVID-19 symptoms on patients’ lives.

## Figures and Tables

**Figure 1 ijerph-21-00591-f001:**
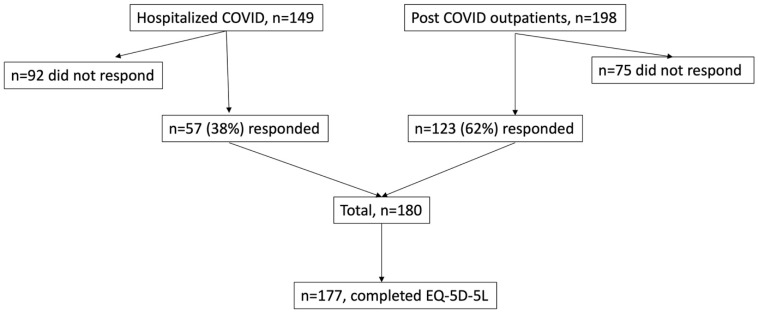
The flow chart of the response rate.

**Figure 2 ijerph-21-00591-f002:**
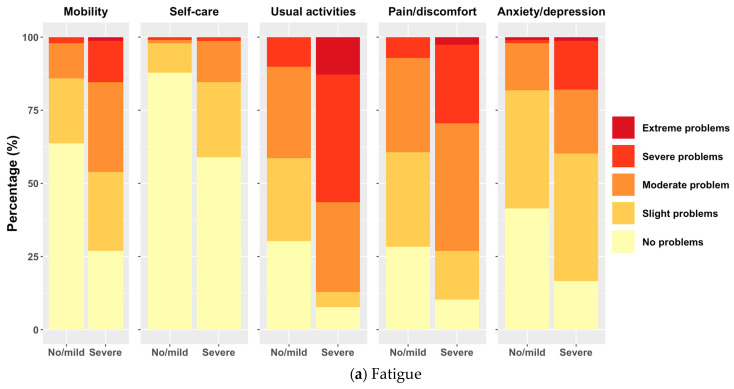
Distribution of study patients on EQ-5D-5L dimensions for different severity levels of fatigue, memory and concentration problems, as well as dyspnea.

**Table 1 ijerph-21-00591-t001:** Characteristics of the whole study population (n = 177) and subgroups of patients with no/milder and severe persistent symptoms, such as fatigue, memory/concentration problems and dyspnea.

	All Patients	Fatigue Severity		Memory/Concentration Problems Severity		Dyspnea Severity	
		No/Milder (0–6)	Severe (7–10)	No/Milder (0–6)	Severe (7–10)	No/Milder (0–6)	Severe (7–10)
Age, mean (SD)	52.2 (11.5)	53.7 (10.6)	50.2 (12.2)	53.8 (10.6)	50.0 (12.3)	52.3 (11.5)	51.7 (11.2)
Female, n (%)	98 (56)	49 (51)	49 (63)	52 (51)	46 (62)	85 (55)	13 (65)
Country of birth, n, (%) Sweden	144 (82)	74 (76)	70 (90)	78 (77)	66 (89)	125 (81)	19 (95)
Education level, n (%) Up to secondary school Vocational education University	80 (47) 27 (16) 64 (37)	42 (43) 16 (16) 39 (40)	38 (51) 11 (15) 25 (33)	47 (46) 18 (18) 36 (35)	33 (47) 9 (13) 28 (40)	73 (48) 24 (16) 54 (36)	7 (35) 3 (15) 10 (50)
Working status, n (%) Working Unemployed On sick leave Retired Student Other	119 (68) 5 (3) 30 (17) 13 (7) 6 (3) 3 (2)	74 (75) 3 (3) 8 (8) 8 (8) 6 (6) 0	45 (58) 2 (3) 22 (29) 5 (6) 0 3 (4)	73 (72) 1 (1) 12 (12) 10 (10) 5 (5) 1 (1)	46 (62) 4 (5) 18 (24) 3 (4) 1 (1) 2 (3)	107 (69) 4 (3) 25 (16) 13 (8) 4 (3) 3 (2)	12 (60) 1 (5) 5 (25) 0 2 (10) 0
Marital status, n (%) Married Living together Divorced Widower Single	95 (54) 42 (24) 15 (8) 1 (1) 24 (14)	60 (61) 17 (17) 10 (10) 1 (1) 11 (11)	35 (45) 25 (32) 5 (6) 0 13 (17)	59 (57) 22 (21) 10 (10) 1 (1) 11 (11)	36 (49) 20 (27) 5 (7) 0 13 (18)	81 (52) 39 (25) 14 (9) 1 (1) 22 (14)	14 (70) 3 (15) 1 (5) 0 2 (10)
Smoking, n (%) Never smoked Ex-smoker Current smoker	117 (66) 54 (31) 5 (3)	67 (68) 28 (29) 3 (3)	50 (64) 26 (33) 2 (3)	69 (68) 30 (29) 3 (3)	48 (65) 24 (32) 2 (3)	103 (66) 49 (31) 4 (3)	14 (70) 5 (25) 1 (5)
Hospitalized, n (%)	86 (50)	62 (64)	24 (32)	60 (61)	26 (36)	80 (53)	6 (30)
Hypertension Heart disease Hypo/hyperthyroidism Diabetes mellitus Lung disease Cancer Immunosuppressive treatment Depression/Anxiety Chronic pain	62 (35) 12 (7) 18 (10) 14 (8) 48 (27) 6 (3) 8 (5) 64 (36) 38 (21)	35 (35) 6 (6) 8 (8) 10 (10) 26 (26) 4 (4) 5 (5) 28 (28) 16 (16)	27 (35) 6 (8) 10 (13) 4 (5) 22 (28) 2 (3) 3 (4) 36 (46) 22 (28)	34 (33) 8 (8) 10 (10) 10 (10) 26 (25) 6 (6) 5 (5) 24 (23) 15 (15)	28 (38) 4 (5) 8 (11) 4 (5) 22 (30) 0 3 (4) 40 (54) 23 (31)	55 (35) 8 (5) 14 (9) 12 (8) 42 (27) 6 (4) 7 (4) 56 (36) 33 (21)	7 (35) 4 (20) 4 (20) 2 (10) 6 (30) 0 1 (5) 8 (40) 5 (25)
BMI, mean (SD)	28.9 (5.7)	28.9 (5.3)	28.9 (6.2)	28.5 (5)	29.4 (6.5)	28.9 (5.3)	28.9 (8.5)
Symptom severity at onset, median (IQR)	4 (1)	4 (1)	4 (1)	4 (1)	4 (1)	4 (1)	4 (1)
Mean number of persistent symptoms, mean (SD)	2.3 (1.8)	1.2 (1.1)	3.6 (1.6)	1.4 (1.4)	3.4 (1.7)	2.0 (1.6)	4.1 (2.0)
EQ-VAS, median (IQR)	5 (4)	6 (3)	3 (4)	6 (3)	3.5 (4)	6 (4)	3 (5)
The best health EQ-5D-5L (11111), n (%)	12 (7)	11 (11)	1 (1)	12 (12)	0	12 (8)	0
Utility score EQ-5D-5L, mean (SD)	0.77 (0.22)	0.87 (0.14)	0.65 (0.25)	0.86 (0.17)	0.66 (0.24)	0.8 (0.22)	0.61 (0.23)

**Table 2 ijerph-21-00591-t002:** Spearman’s rank correlation of EQ-5D-5L dimensions and utility scores with fatigue, memory/concentration problems, and dyspnea.

	Mobility	Self-Care	Usual Activities	Pain/ Discomfort	Anxiety/ Depression	Dyspnea	Memory/ Concentration Problems	Utility Score
Fatigue	0.46 **	0.37 **	0.65 **	0.40 **	0.39 **	0.47 **	0.75 **	−0.62 **
Memory/ concentration problems	0.35 **	0.27 **	0.72 **	0.36 **	0.45 **	0.29 **	-	−0.65 **
Dyspnea	0.29 **	0.32 **	0.30 **	0.19 *	0.18 *	-	-	−0.33 **

Note: * Statistically significant at a 5% level (*p* < 0.05); ** Statistically significant at a 1% level (*p* < 0.01).

**Table 3 ijerph-21-00591-t003:** Multiple regression analysis of EQ-5D-5L for fatigue, memory/concentration problems and dyspnea.

Fatigue		Memory/Concentration Problems		Dyspnea	
Independent Variables	Unstandardized Beta (95% CI)	Independent Variables	Unstandardized Beta (95% CI)	Independent Variables	Unstandardized Beta (95% CI)
Intercept	1.76 (0.84, 2.67) **	Intercept	0.99 (0.12, 1.86) *	Intercept	1.48 (0.52, 2.43) **
Mobility level 2	0.07 (−0.88, 1.01)	Mobility level 2	−0.57 (−1.46, 0.33)	Mobility level 2	0.19 (−0.79, 1.18)
Mobility level 3	0.80 (−0.31, 1.91)	Mobility level 3	−0.36 (−1.42, 0.69)	Mobility level 3	0.39 (−0.77, 1.55)
Mobility level 4	−0.08 (−2.02, 1.86)	Mobility level 4	−1.29 (−3.14, 0.56)	Mobility level 4	1.78 (−0.25, 3.81)
Mobility level 5	−1.41 (6.46, 3.64)	Mobility level 5	−4.37 (−9.19, 0.45)	Mobility level 5	3.18 (−0.25, 3.81)
Self-care level 2	0.58 (−0.53, 1.69)	Self-care level 2	0.16 (−0.90, 1.22)	Self-care level 2	1.46 (0.30, 2.62) *
Self-care level 3	1.02 (−0.60, 2.65)	Self-care level 3	0.56 (−0.99, 2.11)	Self-care level 3	0.14 (−1.56, 1.84)
Self-care level 4	−4.54 (−7.99, −1.09) *	Self-care level 4	−3.86 (−7.15, −0.58) *	Self-care level 4	−0.54 (−4.14, 3.07)
Usual activities level 2	1.29 (0.13, 2.44) *	Usual activities level 2	2.14 (1.04, 3.24) **	Usual activities level 2	−0.07 (−1.28, 1.14)
Usual activities level 3	2.72 (1.58, 3.86) **	Usual activities level 3	4.06 (2.98, 5.15) **	Usual activities level 3	0.64 (−0.55, 1.83)
Usual activities level 4	3.95 (2.68, 5.23) **	Usual activities level 4	5.43 (4.21, 6.64) **	Usual activities level 4	1.08 (−0.25, 2.41)
Usual activities level 5	5.65 (3.65, 7.64) **	Usual activities level 5	7.04 (5.13, 8.94) **	Usual activities level 5	1.55 (−0.54, 3.64)
Pain/discomfort level 2	−0.02 (−1.08, 1.04)	Pain/discomfort level 2	0.04 (−0.97, 1.05)	Pain/discomfort level 2	0.52 (−0.58, 1.63)
Pain/discomfort level 3	0.98 (0.00, 1.97)	Pain/discomfort level 3	0.78 (−0.16, 1.72)	Pain/discomfort level 3	0.66 (−0.38, 1.69)
Pain/discomfort level 4	0.95 (−0.49, 2.40)	Pain/discomfort level 4	0.99 (−0.38, 2.37)	Pain/discomfort level 4	−0.73 (−2.24, 0.78)
Pain/discomfort level 5	2.82 (−1.23, 6.87)	Pain/discomfort level 5	3.68 (−0,18, 7.55)	Pain/discomfort level 5	−0.64 (−4.88, 3.59)
Anxiety/depression level 2	0.72 (−0.18, 1.63)	Anxiety/depression level 2	1.01 (0.15, 1.87) *	Anxiety/depression level 2	0.42 (−0.53, 1.37)
Anxiety/depression level 3	0.60 (−0.52, 1.72)	Anxiety/depression level 3	1.12 (0.05, 2.19) *	Anxiety/depression level 3	−0.22 (−1.40, 0.95)
Anxiety/depression level 4	0.75 (−0.94, 2.44)	Anxiety/depression level 4	0.84 (−0.77, 2.44)	Anxiety/depression level 4	1.00 (−0.76, 2.77)
Anxiety/depression level 5	−0.27 (−3.56, 3.03)	Anxiety/depression level 5	−0.91 (−4.05, 2.23)	Anxiety/depression level 5	3.08 (−0.36, 6.52)
Adj R-squared	0.47	Adj R-squared	0.55	Adj R-squared	0.14

Note: * Statistically significant at a 5% level (*p* < 0.05); ** Statistically significant at a 1% level (*p* < 0.01).

**Table 4 ijerph-21-00591-t004:** Explanatory power of EQ-5D-5L with and without fatigue, memory/concentration problems and dyspnea for EQ-VAS.

Dependent Variable	Independent Variables	Adjusted *R*^2^	F Value
EQ VAS	Utility score **	0.376	106.7 **
	MO *-ADL **-AC **-PA *-AN * ^1^	0.577	13.63 **
	MO *-ADL **-AC **-PA-AN-FA ** ^2^	0.632	16.1 **
	MO *-ADL **-AC **-PA-AN-MC * ^3^	0.587	13.48 **
	MO-ADL **-AC **-PA-AN *-DY ^4^	0.585	13.38 **
	MO *-ADL **-AC **-PA-AN-FA **-MC ^5^	0.630	15.26 **
	MO *-ADL **-AC **-PA-AN-FA **-DY ^6^	0.630	15.24 **
	MO *-ADL **-AC **-PA-AN-MC-DY ^7^	0.591	13.1 **
	MO *-ADL **-AC **-PA-AN-MC-DY-FA ** ^8^	0.628	14.48 **

MO Mobility, ADL Self-care, AC Usual activities, PA Pain/discomfort, AN Anxiety/depression, FA Fatigue, MC Memory/concentration problems, DY Dyspnea. * Statistically significant at a 5% level (*p* < 0.05); ** Statistically significant at a 1% level (*p* < 0.01). ^1^ Levels that were significant: MO level 4*; ADL level 3**; AC levels 2*, 3**, 4**, 5**; PA level 3*; AN level 3*. ^2^ Levels that were significant: MO level 4*; ADL level 3**; AC levels 3**, 4**, 5**. ^3^ Levels that were significant: MO level 4*; ADL level 3**; AC levels 3**, 4**, 5**. ^4^ Levels that were significant: ADL level 3**; AC levels 2**, 3**, 4**, 5**; AN level 3*. ^5^ Levels that were significant: MO level 4*; ADL level 3**; AC levels 3**, 4**, 5**. ^6^ Levels that were significant: MO level 4*; ADL level 3**; AC levels 3**, 4**, 5**. ^7^ Levels that were significant: MO level 4*; ADL level 3**; AC levels 3**, 4**, 5**. ^8^ Levels that were significant: MO level 4*; ADL level 3**; AC levels 3**, 4**, 5**.

## Data Availability

Data can be made available on demand (e-mail: marta.kisiel@medsci.uu.se).

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
