# Peer review of "Enhancing EQ-5D-5L Sensitivity in Capturing the Most Common Symptoms in Post-COVID-19 Patients: An Exploratory Cross-Sectional Study with a Focus on Fatigue, Memory/Concentration Problems and Dyspnea Dimensions"

_ijerph, 2024, doi:10.3390/ijerph21050591_

Round 1

Reviewer 1 Report

Comments and Suggestions for Authors

The study investigated the early detection of symptoms such as fatigue, memory/concentration problems and shortness of breath, which are frequently encountered in patients after COVID-19, with the EQ-5D-5L scale. As far as we can examine in the literature, this is the first study in which EQ-5D-5L was used in COVID-19 disease. Therefore, the data obtained may be useful for public health. It is appropriate to accept.

The authors stated that they recorded the study group cases between April and July 2021. Since COVID-19 vaccines were in use at this time, were there any cases in the study group who had been vaccinated or had the infection before?The study investigated the early detection of symptoms such as fatigue, memory/concentration problems and shortness of breath, which are frequently encountered in patients after COVID-19, with the EQ-5D-5L scale. As far as we can examine in the literature, this is the first study in which EQ-5D-5L was used in COVID-19 disease. Therefore, the data obtained may be useful for public health. It is appropriate to accept.

The authors stated that they recorded the study group cases between April and July 2021. Since COVID-19 vaccines were in use at this time, were there any cases in the study group who had been vaccinated or had the infection before?The study investigated the early detection of symptoms such as fatigue, memory/concentration problems and shortness of breath, which are frequently encountered in patients after COVID-19, with the EQ-5D-5L scale. As far as we can examine in the literature, this is the first study in which EQ-5D-5L was used in COVID-19 disease. Therefore, the data obtained may be useful for public health. It is appropriate to accept.

The authors stated that they recorded the study group cases between April and July 2021. Since COVID-19 vaccines were in use at this time, were there any cases in the study group who had been vaccinated or had the infection before?

Comments on the Quality of English Language

Minor modifications are needed.

Author Response

Thank you for the comments that helped us to improve the manuscript!

Below are answers and changes made in the manuscript as a response to these comments.

Apart from this, the manuscript has also undergone a linguistic revision.

Rev 1

The authors stated that they recorded the study group cases between April and July 2021. Since COVID-19 vaccines were in use at this time, were there any cases in the study group who had been vaccinated or had the infection before?

Reply: We do not have information on the vaccination status, and we added this to the limitation section (line 413).

We attachached the revised manuscript. 

Best Regards,

Marta Kisiel and co-authors

Reviewer 2 Report

Comments and Suggestions for Authors

This study revealed that EQ-5D-5L could partly captured fatigue and memory/concentration problems as common persistent symptoms of covid 19, while poorly captured dyspnea as another common persistent symptoms.  

The limitation of laboratory data and also the number of samples influence the grouping of variables and could reduce the sensitivity of EQ-5D-5L if compare to previous studies.

Using EQ-5D-5L for differentiate between hospitalized group and outpatients group especially for fatigue, memory problems and dyspnea perhaps could give another good results. The authors did not perform this analysis.

I agree with the limitation stated by authors which could decreased the sensitivity of EQ-5D-5L.

Author Response

Rev 2 had no questions to the authors.

Reviewer 3 Report

Comments and Suggestions for Authors

Dear Authors,

Go thro' my suggestions to improve the paper.

Abstract:

To improve the abstract of the study on the EQ-5D-5L tool's ability to assess post-COVID-19 symptoms, start with the research objective and emphasize the tool's ability to assess persistent symptoms like fatigue, memory/concentration problems, and dyspnea. It should summarize the important findings, including dyspnea capture constraints. The abstract should also explain how adding fatigue to the EQ-5D-5L evaluation improves HRQoL prediction. Finally, it should recommend improving the EQ-5D-5L to better capture a larger range of post-COVID-19 symptoms, especially underrepresented ones, to better assess HRQoL in these individuals.

Introduction:

The study on the EQ-5D-5L tool's effectiveness in capturing post-COVID-19 conditions should begin by succinctly defining COVID-19 and its global impact, then introduce the WHO's post-COVID-19 condition and its prevalence and long-term effects on patients. It's important to briefly detail this condition's prevalent symptoms, such as fatigue, dyspnea, and cognitive difficulties, and their influence on quality of life. Next, introduce the EQ-5D-5L measure for assessing health-related quality of life, acknowledging its broad use and the research gap on its ability to capture all post-COVID-19 symptoms. End the paragraph by expressing the research goal: to determine if the EQ-5D-5L tool can accurately assess health-related quality of life by capturing post-COVID-19 symptoms and how adding them may improve its predictive accuracy. It gives a concise overview, sets the study's context, and directly addresses the research goals.

Materails and Methods :

The "COMBAT post Covid" initiative at Uppsala University Hospital sought to capture the long-term effects of COVID-19. What criteria were used to distinguish between the "Hospitalized COVID" and "Post COVID outpatients" cohorts?

Why was a questionnaire used to collect data, and how did the "Hospitalized COVID" and "Post COVID outpatients" cohorts compare in response rates?

How were post-COVID-19 patients' health-related quality of life (HRQoL) assessed using the EQ-5D-5L dimensions and EQ Visual Analogue Scale (EQ-VAS), and what process was employed to create a health profile?

Given the wide range of persistent symptoms reported by patients, how were fatigue, memory/concentration issues, and dyspnea prioritized for additional research, including the 7-symptom severity cut-off?

How were Spearman's correlation and multiple regression analysis used to assess the relationship between EQ-5D-5L dimension scores and post-COVID-19 symptom severity, and how were multicollinearity in the statistical models avoided?

Results:

How did the study population's mean EQ-5D-5L utility score correspond with persistent symptoms and self-reported HRQoL as measured by EQ-VAS scores?

Why was a cut-off point of 7 adopted to distinguish between no/mild and severe persistent symptom severity, and how did this classification affect the analysis of symptom severity and health outcomes?

What were the strongest EQ-5D-5L correlations for fatigue, memory/concentration, and dyspnea patients, and how did they relate to utility scores?

Multiple regression analyses showed that some EQ-5D-5L dimensions strongly affected fatigue, memory/concentration, and dyspnea. Could you explain how these dimensions helped you evaluate symptom severity?

The EQ-VAS's explanatory power increased when fatigue was added to the EQ-5D-5L dimensions. How big was this improvement, and what does it say about the potential benefits of altering the EQ-5D-5L to better capture post-COVID-19 health outcomes?

Discussion :

Separate the discussion into theoretical and managerial impications of your work.

Conclusion :

To support the study's conclusion that the EQ-5D-5L tool is effective in assessing health-related quality of life (HRQoL) in post-COVID-19 patients, it should succinctly summarize the key findings, emphasizing the tool's partial success in capturing fatigue and memory/concentration issues and its limitations with dyspnea The recommendation to add a tiredness dimension should be presented as a strategic upgrade to boost the tool's sensitivity to post-COVID-19 situations. The conclusion should recommend more study to improve the EQ-5D-5L to better assess the multidimensional effects of post-COVID-19 symptoms on patients' life. A stronger close will result from focusing on both the findings and their implications for future research and practice.

Author Response

Thank you for the comments that helped us to improve the manuscript!

Below are answers and changes made in the manuscript as a response to these comments.

Apart from this, the manuscript has also undergone a linguistic revision.

Rev 3

Abstract:

To improve the abstract of the study on the EQ-5D-5L tool's ability to assess post-COVID-19 symptoms, start with the research objective and emphasize the tool's ability to assess persistent symptoms like fatigue, memory/concentration problems, and dyspnea. It should summarize the important findings, including dyspnea capture constraints. The abstract should also explain how adding fatigue to the EQ-5D-5L evaluation improves HRQoL prediction. Finally, it should recommend improving the EQ-5D-5L to better capture a larger range of post-COVID-19 symptoms, especially underrepresented ones, to better assess HRQoL in these individuals.

Reply: We agree with the reviewer's view on how our abstract should be organized. We tried to do this already in our first version but have added some changes in the new version of the abstract.

Introduction:

The study on the EQ-5D-5L tool's effectiveness in capturing post-COVID-19 conditions should begin by succinctly defining COVID-19 and its global impact, then introduce the WHO's post-COVID-19 condition and its prevalence and long-term effects on patients.

-It's important to briefly detail this condition's prevalent symptoms, such as fatigue, dyspnea, and cognitive difficulties, and their influence on quality of life.

-Next, introduce the EQ-5D-5L measure for assessing health-related quality of life, acknowledging its broad use and the research gap on its ability to capture all post-COVID-19 symptoms.

-End the paragraph by expressing the research goal: to determine if the EQ-5D-5L tool can accurately assess health-related quality of life by capturing post-COVID-19 symptoms and how adding them may improve its predictive accuracy. It gives a concise overview, sets the study's context, and directly addresses the research goals.

Reply: We have now tried to structure the introduction in the way the reviewer suggested.

Materails and Methods :

The "COMBAT post Covid" initiative at Uppsala University Hospital sought to capture the long-term effects of COVID-19. What criteria were used to distinguish between the "Hospitalized COVID" and "Post-COVID outpatients" cohorts?

Reply: Thank you for the comment. The two groups, Hospitalized COVID and Post COVID, are two cohorts collected separately. Hospitalized COVID comprised patients after COVID-19 who require hospitalization one year prior to the follow-up, whereas the post-COVID outpatients group was collected at the primary care outpatients settled for patients with long-term persistent symptoms after COVID-19. Both cohorts were collected in the same Swedish city, Uppsala. For details, see lines 102-180.

Why was a questionnaire used to collect data, and how did the "Hospitalized COVID" and "Post COVID outpatients" cohorts compare in response rates?

Reply: We used a questionnaire to collect data as we wanted to collect self-reported information on symptoms and clinical and sociodemographic factors in both cohorts. The response rate was 38% in the Hospitalized COVID group and 62% in the Post COVID outpatients group. The survey was performed in different settings, which is the limitation of this study, in addition to the low response rate. This is added to the limitation section, lines 619-621. This study is of exploratory character.

How were post-COVID-19 patients' health-related quality of life (HRQoL) assessed using the EQ-5D-5L dimensions and EQ Visual Analogue Scale (EQ-VAS), and what process was employed to create a health profile?

Reply: Thank you for this comment. In the method section, we described that EQ5D5L was assessed by considering independent dimensions (methods, lines 186-193) or by creating health states converted into a single utility score using a scoring algorithm. This study used the Swedish value set and scoring algorithm to calculate utility scores, lines 249-252, and the EQ Visual Analogue Scale (EQ-VAS), line 182.

Given the wide range of persistent symptoms reported by patients, how were fatigue, memory/concentration issues, and dyspnea prioritized for additional research, including the 7-symptom severity cutoff?

Reply: Thank you for this comment. Patients were asked to rate the intensity of each symptom on a scale from 1 (very mild) to 10 (most severe). A score of zero was assigned if the patient did not experience any symptoms. Based on previous studies on symptoms such as pain, fatigue, and dyspnea graded on a 10-point scale, we set the optimal cutoff point to distinguish between no/mild and severe symptom severity at 7, lines 201-215. However, as we stated in the limitation section Fourth, the methodology employed a simple question to assess persistent symptoms and rate their severity on a 10-point scale, with a cutoff at 7, drawing inspiration from other scales designed for dyspnea and pain.

How were Spearman's correlation and multiple regression analysis used to assess the relationship between EQ-5D-5L dimension scores and post-COVID-19 symptom severity, and how were multicollinearity in the statistical models avoided?

Reply: Thank you for this comment. Spearman's correlation was used to investigate correlations among fatigue, memory/concentration problems, dyspnea, EQ-5D-5L dimensions, and the utility score. Correlation coefficients were interpreted as the following: 0.1–0.29 (poor), 0.3–0.5 (fair), 0.6-0.79 (moderately strong), and 0.8-1.0 (very strong), line 180-183. Multiple regression analysis was used to assess the impact of EQ-5D-5L dimension scores (independent variables) on the severity of the symptoms: fatigue, memory/concentration problems and dyspnea (dependent variables). An exploratory analysis involved nine distinct multiple regression models to compare adjusted R-squared values for EQ-VAS (dependent variable), lines 284-288. Then, we calculated Variance Inflation Factor (VIF) values and found a low VIF value that excluded significant multicollinearity concerns in the subsequent statistical models, Figure S2 and lines 257-282.

Results:

How did the study population's mean EQ-5D-5L utility score correspond with persistent symptoms and self-reported HRQoL as measured by EQ-VAS scores?

Reply. Thank you for the comment. We explained it in lines 320-327 and Tabel 2, as well as lines 404-412.

Why was a cutoff point of 7 adopted to distinguish between no/mild and severe persistent symptom severity, and how did this classification affect the analysis of symptom severity and health outcomes?

Reply: Thank you for the comment. Patients were asked to rate the intensity of each symptom on a scale from 1 (very mild) to 10 (most severe). A score of zero was assigned if the patient did not experience any symptoms. Based on previous studies on symptoms such as pain, fatigue, and dyspnea graded on a 10-point scale, we set the optimal cutoff point to distinguish between no/mild and severe symptom severity at 7, lines 201-215. However, as we stated in the limitation section Fourth, the methodology employed a simple question to assess persistent symptoms and rate their severity on a 10-point scale, with a cutoff at 7, drawing inspiration from other scales designed for dyspnea and pain.

What were the strongest EQ-5D-5L correlations for fatigue, memory/concentration, and dyspnea patients, and how did they relate to utility scores?

Reply. Thank you for the comment. We explained this in lines 320-327. and Table 2.

Multiple regression analyses showed that some EQ-5D-5L dimensions strongly affected fatigue, memory/concentration, and dyspnea. Could you explain how these dimensions helped you evaluate symptom severity?

Reply. Thank you for the comment. We explained this in lines 370-385.

The EQ-VAS's explanatory power increased when fatigue was added to the EQ-5D-5L dimensions. How big was this improvement, and what does it say about the potential benefits of altering the EQ-5D-5L to better capture post-COVID-19 health outcomes?

Reply. Thank you for the comment. We explained this in lines 405-411.

Discussion :

Separate the discussion into theoretical and managerial impications of your work.

Reply. Thank you for the comment. We have tried to cover both aspects in a structured way.

Conclusion :

To support the study's conclusion that the EQ-5D-5L tool is effective in assessing health-related quality of life (HRQoL) in post-COVID-19 patients, it should succinctly summarize the key findings, emphasizing the tool's partial success in capturing fatigue and memory/concentration issues and its limitations with dyspnea The recommendation to add a tiredness dimension should be presented as a strategic upgrade to boost the tool's sensitivity to post-COVID-19 situations. The conclusion should recommend more study to improve the EQ-5D-5L to better assess the multidimensional effects of post-COVID-19 symptoms on patients' life. A stronger close will result from focusing on both the findings and their implications for future research and practice.

Reply: We agree and have modified our conclusion.

We attached the revised manuscript. 

Best Regards,

Marta Kisiel and co-authors